# NORMFORMER: IMPROVED TRANSFORMER PRETRAINING WITH EXTRA NORMALIZATION

## ABSTRACT

During pretraining, the Pre-LayerNorm transformer suffers from a gradient magnitude mismatch: gradients at early layers are much larger than at later layers. These issues can be alleviated by our proposed NormFormer architecture, which adds three normalization operations to each layer: a Layer Norm after self attention, head-wise scaling of self-attention outputs, and a Layer Norm after the first fully connected layer. The extra operations incur negligible compute cost (+0.4% parameter increase), but improve pretraining perplexity and downstream task performance for both causal and masked language models ranging from 125 Million to 2.7 Billion parameters. For causal language modeling, adding NormFormer on top of our strongest 1.3B parameter baseline achieves equal performance with 79% as much compute, and matches GPT-3 Large performance in 62% of the cost. In the same compute budget, NormFormer comfortably beats both baselines in both pre-training perplexity and zero shot performance. For masked language modeling, NormFormer improves fine-tuned GLUE performance by 1.9% on average. Code to train NormFormer models is available in REDACTED.

## 1 INTRODUCTION

The original transformer architecture (Vaswani et al., 2017) applies Layer Normalization (Ba et al., 2016) after each sublayer's residual connection ("Post-LN") in order to reduce the variance of the inputs to the following sublayer, i.e.:

$$\text{PostLN}(x) = \text{LayerNorm}(x + \text{Sublayer}(x)),$$

with

$$\text{LayerNorm}(x) = \frac{x - E[x]}{\sqrt{Var[x] + \epsilon}} \cdot \gamma + \beta,$$

where $\gamma$ and $\beta$ are trainable parameters, and $\epsilon$ is a small constant. Recent work has shown empirically and theoretically that Post-LN transformers tend to have larger magnitude gradients in later layers compared to earlier layers (Xiong et al., 2020) and has advocated moving the LayerNorm operation to the beginning of each sublayer ("Pre-LN"; see Figure 1, left), i.e.:

$$\text{PreLN}(x) = x + \text{Sublayer}(\text{LayerNorm}(x)).$$

In practice Pre-LN transformers can be trained with larger learning rates, shorter learning rate warmup and often yield improved performance compared to Post-LN transformers (Xiong et al., 2020), so most recent, large pretrained language models tend to use Pre-LN transformers (Baevski & Auli, 2019; Radford et al., 2019; Raffel et al., 2020; Brown et al., 2020; Lieber et al., 2021). In this work we show that, while Pre-LN improves stability over Post-LN, it has the opposite side effect: gradients at earlier layers tend to be larger than gradients at later layers, thereby limiting the learning rate.[1] We propose `NormFormer`, which alleviates the gradient magnitude mismatch by adding 3 normalization operations to each layer (see Figure 1, middle). These operations reduce gradients to early layers and increase gradients to later layers, bringing their magnitudes closer together.

---

[1]Intuitively, training stably requires that the largest weight update not be too large, while training efficiently requires large weight updates.

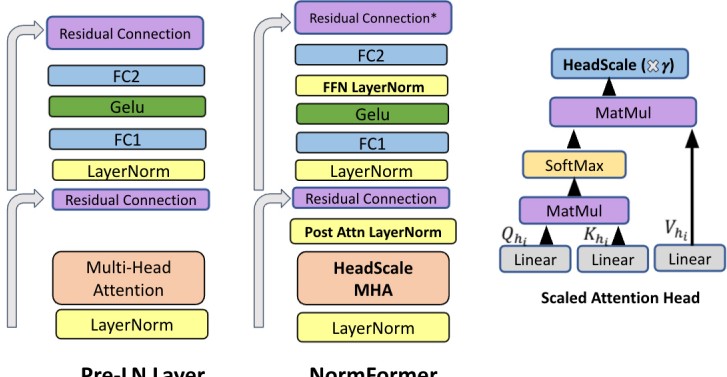

Figure 1: Left: a baseline Pre-LayerNorm transformer layer. Center: `NormFormer`, with the three proposed additions in bold. Right: a single attention head with our proposed `HeadScale` operation applied prior to the output projection with trainable parameters $\gamma_i$. * When applied, residual scaling impacts the second residual connection in each layer.

Compared to compute-matched, well-tuned Pre-LN baselines, `NormFormer` models reach target pretraining perplexities faster and achieve better pretraining perplexities and downstream task performance.

The rest of this paper is organized as follows: Section 2 describes the proposed modifications. Section 3 describes related work. Section 5 shows pretraining and downstream task performance for fully trained `NormFormer` models against well-tuned, compute-matched baselines. Section 6 shows the gradient mismatch introduced by Pre-LN and how `NormFormer` alleviates it. Section 6.1 analyzes residual scaling, a related technique proposed to stabilize Post-LN architectures (Xiong et al., 2020; Zhu et al., 2021). Section 7 shows that removing any of the added operations degrades performance and that `NormFormer` improves over the baseline at a wide range of hyperparameter configurations. Section 9.1 compares `NormFormer` to Related Work from other domains.

## 2 APPROACH

### 2.1 NORMFORMER

`NormFormer` includes three modifications to the Pre-LN transformer: First, we apply head-wise scaling inside the attention module and add two additional LayerNorm operations: one after the attention module and a second after the first fully connected layer. The modifications introduce a small number of additional learnable parameters, which provide a cost-effective way for each layer to change the magnitude of its features, and therefore the magnitude of the gradients to subsequent components. The changes are visualized in Figure 1 and described below.

**Scaling Attention Heads**   The standard multi-head attention operation is defined as:

$$\text{MultiHeadAttention}(Q, K, V) = \text{Concat}(\text{h}_1, \ldots, \text{h}_n)W^O$$
$$\text{h}_i = \text{Attention}(QW_i^Q, KW_i^K, VW_i^V)$$
$$\text{Attention}(Q, K, V) = \text{softmax}\left(\frac{QK^T}{\sqrt{d_k}}\right)V,$$

where $n$ is the number of heads, $i$ is the attention head index, $d_k$ is the dimensionality of the keys and $W^O, W_i^Q, W_i^K, W_i^V$ are learned projection matrices for the output, query, key and value, respectively.

We propose scaling the output of each attention head via learned scalar coefficients $\gamma_i$:

$$\text{HeadScaleMHA}(Q, K, V) = \text{Concat}(\gamma_1 \text{h}_1, \ldots, \gamma_n \text{h}_n)W^O$$

where $\gamma$ are learnable parameters initialized to 1.

**Additional Layer Normalization and Putting it All Together**   In the Pre-LN transformer each layer $l$ modifies an input $x_l$ as follows:

$$x_{l+1}^{\text{PreLN}} = \text{FFN}(\text{MHA}(x_l))$$

$$\text{where} \quad \text{MHA}(x) = x + \text{MultiHeadAttention}(\text{LN}(x), \text{LN}(x), \text{LN}(x))$$
$$\text{FFN}(x) = x + \sigma(\text{LN}(x)W_1 + b_1)W_2 + b_2$$
$$\text{LN}(x) = \text{LayerNorm}(x)$$

In this work $\sigma$ is the GELU non-linear activation introduced in Hendrycks & Gimpel (2016).

Our overall method, `NormFormer`, instead modifies each input $x_l$ as:

$$x_{l+1}^{\text{NormFormer}} = \text{NormFFN}(\text{NormScaledMHA}(x_l))$$

$$\text{where} \quad \text{NormScaledMHA}(x) = x + \textbf{LN}(\textbf{HeadScaleMHA}(\text{LN}(x), \text{LN}(x), \text{LN}(x)))$$
$$\text{NormFFN}(x) = \boldsymbol{x} + \textbf{LN}(\sigma(\text{LN}(x)W_1 + b_1))W_2 + b_2$$

where bolded operations are newly introduced.

## 3   RELATED WORK

**Architectural Modifications**   GradInit (Zhu et al., 2021) introduces a set of scalars and biases for initialization based on a variance heuristic, and Admin (Liu et al., 2020) applies a similar heuristic in profiling and initialization stages. These works also use variants of our `ResScale` operation, which we find helpful at small scale and harmful at large scale. Our approach, in contrast, only has new learnable parameters without variance heuristics, and has no extra stages or changes in initialization.

Shazeer (2020) proposes FFN-GeGLU, which includes scaling but no normalization, in the same position as our FFN LN. Ding et al. (2021) propose related stabilization strategies for text to image generation tasks with larger models including a down-scaled embedding gradient, a slightly different LN formulation, LN after the final fully connected layer, and the same post-attention LN. Section 9.1 compares `NormFormer` to these proposals, as well as the T5 LayerNorm Variant (Raffel et al., 2020), which removes the bias and the mean subtraction from the normalization.

Our `HeadScale` operation is related to that used in Chen et al. (2021), but used differently. Whereas that work prunes attention heads with low $\gamma$ parameters, we use the $\gamma$ parameters to improve pretraining performance.

Press et al. (2020a) proposes an architecture where instead of interleaving attention and feed forward sublayers, the attention all happens first. This increases the number of late FFN parameters, rather than increasing their importance and gradient norm, as our FFN LN does, and does not impact stability.

**Residual Scaling**   Standard Post-LN transformers simply sum the previous output (residual) with the new output. Recent work attempts to stabilize transformers by weighting the residual connection for each layer (Zhu et al., 2021; Liu et al., 2020; Touvron et al., 2021). We thus experiment with scaling the residual in each embedding dimension via learned scalar coefficients $(\lambda_{resid})_i$:

$$\text{ResScale}(\text{x}) = \lambda_{resid} \circ x + \text{Sublayer}(\text{LayerNorm}(x))$$

where $\circ$ is elementwise multiplication, and $\lambda_{resid}$ are learned parameters initialized to 1.

While this can be applied at any normalization layer, we find it it most effective for normalizing the feedforward network (FFN) submodule for the smaller sized language models. In this setting,

$$\text{NormFFN}(x) = \boldsymbol{\lambda_{resid}} \circ \boldsymbol{x} + \textbf{LN}(\sigma(\text{LN}(x)W_1 + b_1))W_2 + b_2$$

| Model Size | GPT-3 Paper | Baseline | NormFormer |
|---|---|---|---|
| 125M | 6e-4 | 3e-3 | 3e-3 |
| 355M | 3e-4 | 1e-3 | 1e-3 |
| 1.3B | 2e-4 | 6e-4 | 6e-4 |

Table 1: Searching for learning rates on our dataset results in higher values than reported in Brown et al. (2020), providing stronger baselines to compare to our NormFormer architecture.

For 1.3B parameter models and larger, scaling residuals hurts performance (see discussion in Section 6.1), so `ResScale` is not used in our 1.3B and 2.7B CLM results. Additionally, we experiment with initializing $\lambda_{resid} = 1e - 5$, following (Touvron et al., 2021), as well replacing addition in residual connections with concatenation (Davis et al., 2021) in Section 9.1.

# 4 EXPERIMENTS

**Causal Language Models**   We pretrain causal LMs (CLM) that roughly match the "Small" (125M parameter), "Medium" (355M), "Large" (1.3B) and "XL" (2.7B) sizes from Brown et al. (2020).

Our model architecture differs from Brown et al. (2020) in two ways: (1) we use only dense attention, while they alternate between dense and locally banded sparse attention; (2) we train our models with sinusoidal positional embeddings, following Shortformer (Press et al., 2020b), since early experiments found this to produce comparable results with fewer learned parameters.

We train the baseline models for 300 billion tokens. We train `NormFormer` models for an equivalent number of GPU hours, which typically results in 2-6% fewer steps and tokens due to the additional overhead of the normalization operations.

On our dataset, we find that the learning rates proposed in GPT-3 are suboptimally low.[2] For both baseline and NormFormer at each size besides 2.7B, we tune the learning rate by training models for 50,000 steps and selecting the best performing learning rate among: $\{1e-4, 6e-4, 3e-4, 6e-4, 1e-3, 3e-3\}$. The learning rates we obtained from this process, shown in Table 1, are 3-5 times larger than those used in the GPT-3 paper. Additionally, we have verified that the baseline and NormFormer both perform worse at the full training budget with the GPT-3 learning rates than with the higher learning rates. Other hyperparameters do not differ from GPT-3.[3]

**Large scale experiments**   We also train three large-scale models with 2.7B parameters. Our first baseline is a replicated version of GPT-3-2.7B with GELU activations, the published learning rate (1.6e-4) and the same number of training steps and tokens (286K steps; 300B tokens). This model slightly exceeds the reference zero shot performance (Brown et al., 2020). Next, we train two variants of GPT3-2.7B with $Relu^2$ activations (So et al., 2021), but use slightly fewer training steps (20% less) for compute efficiency. The first of these uses the baseline learning rate (1.6e-4) and the second uses `NormFormer-2.7B` with a higher learning rate of 6e-4. We note that training baseline 2.7B CLMs (i.e., without `NormFormer` modifications) with a higher 6e-4 learning rate diverged and failed to train. However, as opposed to the smaller architectures, we did not exhaustively tune the learning rate, so it is possible that an intermediate value would perform better.

**Zero Shot Evaluation**   In addition to validation perplexity, we evaluate CLMs on a subset of the tasks that GPT3 evaluated on in a zero-shot setting (Brown et al., 2020), with the same prompts. We select WinoGrande (Sakaguchi et al., 2020), StoryCloze (Mostafazadeh et al., 2016), OpenBookQA (Mihaylov et al., 2018), HellaSwag (Zellers et al., 2019) and PIQA (Bisk et al., 2020) because GPT3 showed strong performance on these tasks at small scale, as well as consistently improving performance with scale.

---

[2]The difference in optimal learning rates may be due partly to architectural differences between our baseline and GPT-3 (e.g., not using locally banded sparse attention).

[3]See Table 2.1 in Brown et al. (2020).

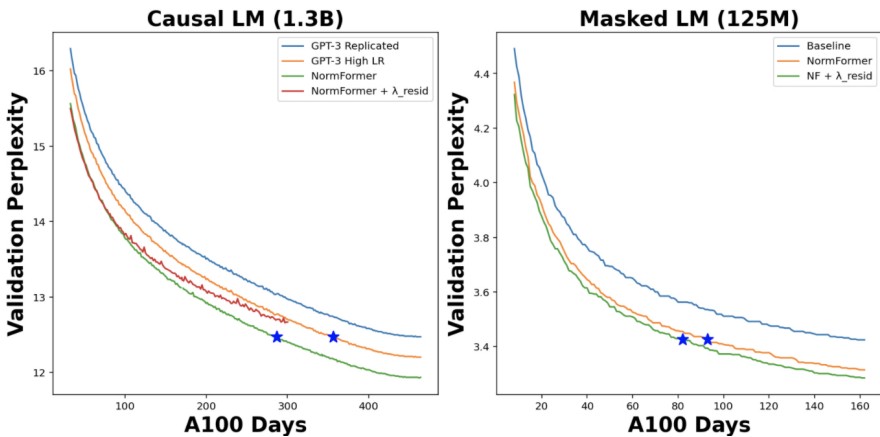

Figure 2: Pretraining perplexity on held-out validation data for Causal and Masked Language Models as a function of training compute (GPU days). The blue stars show the point where a model matches the baseline's lowest perplexity.

**Masked Language Models (MLM)**   We adopt the RoBERTa-base, Pre-LN architecture and hyperparameters used in Liu et al. (2019). For the baseline, we pretrain for 2 million batches of 1 million tokens, about $\frac{1}{4}$ of the training budget of the original `roberta-base`. NormFormer runs through 1.92 million batches in the same amount of time.

**Fine-Tuning**   We fine-tune both the baseline MLM and NormFormer with learning rates $1e{-}5, 1e{-}4, 3e{-}4, 1e{-}3, 3e{-}3, 6e{-}3$ and report the best performance on the validation set for each GLUE task (Wang et al., 2019), following Liu et al. (2019). Other fine-tuning hyperparameters match those used for `roberta-base` in Liu et al. (2019).

**Pretraining data**   We pretrain all models on a collection of English language text including the English portion of the CC100 corpus (Conneau et al., 2020) as well as the data from Liu et al. (2019), consisting of BookCorpus (Zhu et al., 2019), English Wikipedia and filtered subsets of Common Crawl. We encode our data with the byte-level Byte Pair Encoding (BPE) vocabulary from Liu et al. (2019), originally introduced in Radford et al. (2019). The combined dataset contains around 450GB of uncompressed text and 110B BPE tokens. We hold out 40M BPE tokens from this data as a validation set on which we report pretraining perplexities.

**Implementation details**   We train our causal and masked language models in `fairseq` (Ott et al., 2019; Paszke et al., 2019). Although NormFormer introduces fewer than 0.07% additional parameters, it slows individual training updates and increases memory usage between 2% (2.7B model) to 6% (125M model) due to the FFN LNs. Accordingly, we compare NormFormer to baseline models trained for an equal amount of GPU time, i.e., controlling for compute rather than the number of training updates. Finally, we note that the `HeadScale` operation can be moved outside the self attention module to allow the use of the very efficient pytorch `F.multihead_attention`. This change reduces overhead without noticeable performance degradation.

## 5   RESULTS

We report pretraining perplexities for CLMs and MLMs as a function of training wall-time (GPU days) in Figure 2. We observe that NormFormer trains significantly faster and achieves better validation perplexities for a given training compute budget. The blue stars mark the first validation step where NormFormer matches the baseline's lowest perplexity and shows that NormFormer matches Pre-LN models while needing only 60% and 57% as much compute for CLM and MLM models, respectively. This is particularly impressive since NormFormer models take 2-6% longer for each training step and thus see less data than Pre-LN models in this comparison. The left side blue line in Figure 2 shows the failed attempt to add `ResScale` to `NormFormer-1.3B`.

| | $|\theta|$ | LR | $Relu^2$ | $\lambda_{resid}$ | Steps | PPL | HS | PI | WG | SC | OB | Avg |
|---|---|---|---|---|---|---|---|---|---|---|---|---|
| Random Baseline | - | - | - | - | - | - | 25.0 | 50.0 | 50.0 | 50.0 | 25.0 | 40.0 |
| GPT3-125M (paper) | 124.4 | 6e-4 | - | - | 572K | - | 33.7 | 64.6 | 52.0 | 63.3 | 35.6 | 49.8 |
| GPT3-125M (replicated) | 124.4 | 6e-4 | - | - | 572K | 21.11 | 33.7 | 66.5 | 52.2 | 66.1 | 35.4 | 50.8 |
| GPT3-125M (High LR) | 124.4 | 3e-3 | - | - | 572K | 21.09 | 35.3 | 67.5 | 50.5 | 66.3 | 35.0 | 50.9 |
| NormFormer-125M | 124.5 | 3e-3 | - | - | 540K | 20.34 | 34.9 | 67.1 | 52.3 | 66.3 | 38.0 | 51.7 |
| NormFormer-125M | 124.5 | 3e-3 | - | ✓ | 539K | **20.11** | 34.9 | 65.9 | 53.4 | 67.5 | 40.0 | **52.3** |
| GPT3-355M (paper) | 354.7 | 3e-4 | - | - | 572K | - | 43.6 | 70.2 | 52.1 | 68.5 | 43.2 | 55.5 |
| GPT3-355M (replicated) | 354.7 | 3e-4 | - | - | 572K | 15.41 | 46.1 | 70.8 | 54.6 | 71.1 | 41.2 | 56.8 |
| GPT3-355M (High LR) | 354.7 | 1e-3 | - | - | 572K | 14.85 | 48.4 | 71.7 | 53.8 | 73.3 | 43.4 | 58.1 |
| NormFormer-355M | 355.0 | 1e-3 | - | - | 552K | 14.54 | 49.7 | 71.8 | 56.0 | 73.8 | 43.6 | 59.0 |
| NormFormer-355M | 355.0 | 1e-3 | - | ✓ | 550K | **14.52** | 49.7 | 72.0 | 56.7 | 73.2 | 43.8 | **59.1** |
| GPT3-1.3B (paper) | 1313.5 | 2e-4 | - | - | 286K | - | 54.7 | 75.1 | 58.0 | 73.4 | 46.8 | 61.6 |
| GPT3-1.3B (replicated) | 1313.5 | 2e-4 | - | - | 286K | 12.56 | 58.5 | 74.6 | 58.1 | 76.8 | 49.4 | 63.5 |
| GPT3-1.3B (High LR) | 1313.5 | 6e-4 | - | - | 286K | 12.21 | 57.5 | 74.3 | 59.3 | 76.3 | 50.8 | 63.6 |
| NormFormer-1.3B | 1314.0 | 6e-4 | - | - | 275K | **11.94** | 60.5 | 74.5 | 60.1 | 77.5 | 50.8 | **64.7** |
| GPT3-2.7B (paper) | 2648.7 | 1.6e-4 | - | - | 286K | - | 62.8 | 75.6 | 62.3 | 77.2 | 53.0 | 66.2 |
| GPT3-2.7B (replicated) | 2648.7 | 1.6e-4 | - | - | 286K | 10.92 | 65.9 | 76.6 | 61.4 | 78.2 | 49.6 | 66.3 |
| NormFormer-2.7B | 2649.5 | 6e-4 | ✓ | - | 277K | **10.55** | 68.1 | 78.1 | 64.4 | 79.4 | 53.4 | **68.7** |
| GPT3-2.7B-Relu | 2648.7 | 1.6e-4 | ✓ | - | 230K | 10.99 | 65.9 | 76.1 | 63.2 | 79.3 | 49.4 | 66.8 |
| GPT3-2.7B-Relu | 2648.7 | 6e-4 | ✓ | - | 28K | | | | diverged | | | |
| NormFormer-2.7B | 2649.5 | 6e-4 | ✓ | - | 222K | **10.73** | 67.4 | 77.2 | 64.4 | 78.9 | 52.6 | **68.1** |

Table 2: Zero-Shot Accuracy for Causal LMs for the following tasks: HS: HellaSwag, PI: PIQA, WG: WinoGrande, SC: StoryCloze, OB: OpenBookQA. PPL is validation perplexity during pretraining. *GPT-3 (paper)* results taken from Brown et al. (2020). Horizontal lines group compute-matched runs. *High LR* corresponds to using a larger learning rate than reported in Brown et al. (2020). $\lambda_{resid}$ indicates whether residual scaling was used. $\lambda_{resid}$ did not help at 1.3B scale, as shown in 2, but that run is not compute matched so it is not included here. Model size ($|\theta|$) is reported in millions of parameters.

| | Model Size | $\lambda_{resid}$ | PPL | CoLA | MNLI | MRPC | QNLI | QQP | RTE | SST-2 | Avg |
|---|---|---|---|---|---|---|---|---|---|---|---|
| Baseline | 125.42 | - | 3.42 | 74.3 | 85.9 | 84.6 | 91.6 | 90.7 | 66.4 | 92.9 | 83.77 |
| NormFormer | 125.50 | - | 3.31 | **82.6** | **86.3** | **86.0** | **91.9** | **91.3** | **67.9** | 93.8 | **85.69** |
| NormFormer | 125.51 | ✓ | **3.29** | 80.9 | 86.2 | 85.3 | 91.5 | 91.2 | 62.8 | **94.2** | 84.59 |

Table 3: Masked LM: Pretraining validation perplexity (PPL) and fine-tuned performance on GLUE tasks for Pre-LN and NormFormer models. Note that models are trained for an equal amount of compute, which is less than the publicly-released `roberta-base` models.

We observe a similar trend on downstream tasks. In Table 2 we report zero shot accuracy for causal LMs using the tasks and prompts from Brown et al. (2020). NormFormer outperforms GPT-3 at all sizes. The gains from `Normformer` extra parameters operations outpace the gains from normal scaling laws. Changing the hidden dimension of a 125M parameter model from 768 to 780, for example, results in a 127 million parameter model that is only 0.08 perplexity better than the baseline whereas `NormFormer-125M` adds only 100,000 parameters and is 0.83 perplexity better than the baseline.

For MLM models, we report fine-tuned accuracy on GLUE in Table 3. We again find that Norm-Former MLM models outperform their Pre-LN counterparts on every task (rows 1 vs 2). Adding `ResScale` improves improves pre-training performance marginally (3.29 valid PPL vs 3.31), but the gains to do not translate to finetuned performance.

## 6 ANALYSIS

**Analysis of gradient norms by layer** We begin by examining the magnitude of the gradients at different layers for Post-LN, Pre-LN and NormFormer models, since large magnitude differences in gradients across layers can destabilize training, particularly when training in mixed precision (Micikevicius et al., 2018). Figure 3 shows the average L1 norm of the gradients to the second fully

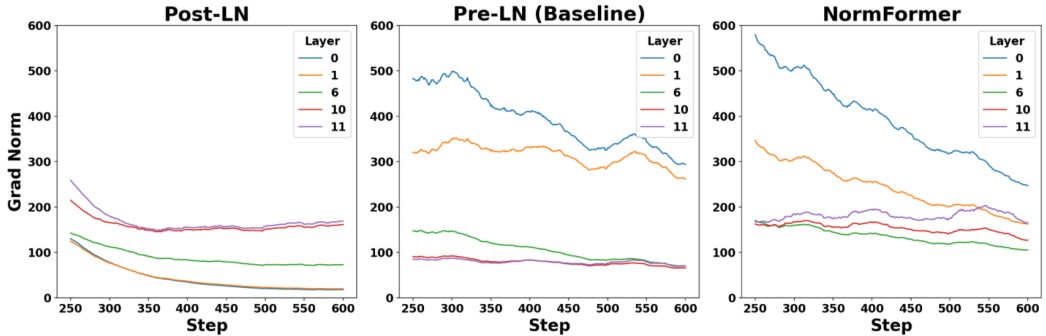

Figure 3: Average L1 norm of gradients to the second fully connected weight for layers 0,1,6,10 and 11, early in training.

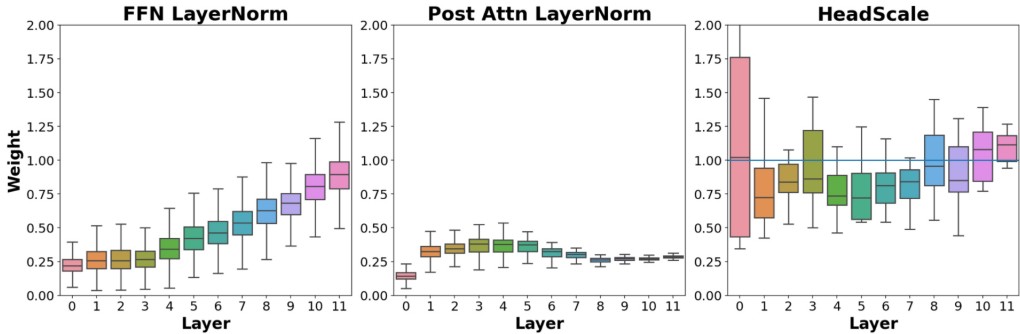

Figure 4: Distribution of learned scaling parameters in three of the added operations. For FFN LN, earlier layers receive downscaled inputs, keeping their gradients in the same range as the gradients of later layers. This plot is discussed in detail in Section 6.

connected weight in various layers for a 12 layer, 125M parameter CLM model at the beginning of training. As reported in past work (Xiong et al., 2020), we observe that the gradients to later layers in Post-LN models are much larger than for earlier layers, and that the gradients to early layers quickly vanish in the early stages of training. Pre-LN models have the opposite behavior, with early layers instead receiving significantly larger gradients than later layers. `NormFormer` brings the average gradient norms closer together for different layers in the network.

In Figure 4 we present the distribution of scaling parameters learned by `NormFormer` models. For the FFN LN, the $\gamma$ parameters are smaller for earlier layers, reducing the magnitude of the inputs to early fully connected parameters, thereby decreasing the magnitude of their gradients. The post attention LN, in the middle of Figure 4, all layers have $\gamma$ coefficients below 1, indicating downscaling.[4] The `HeadScale` $\gamma$ parameters, shown in the rightmost plot in Figure 4 vary more than the others, and have no relationship with depth in the network. We interpret this as evidence that the `HeadScale` parameters dynamically increase the importance of well initialized attention heads, as suggested in Chen et al. (2021).

Reducing gradient mismatch allows training stably with larger learning rates. To measure the stability of an architecture, we train it on a learning rate schedule with a very large peak learning rate, so that the learning rate increases a little each step until the loss explodes. Figure 5 shows that NormFormer models can survive for more updates in this environment than the baseline. For the baseline 125M model (the left most blue dot), the loss eventually explodes, with the activations from multiplying the query and key features at layer 0 overflowing the FP16 range. The down scaling of

---

[4]The downscaling is also apparent in Figure 7 in the Appendix, which plots the change in grad norm for each operation at each layer. It shows that adding extra normalization reduces the gradient norm for all attention parameters at every layer. Only FFN parameters at later layers, have increased gradient norms.

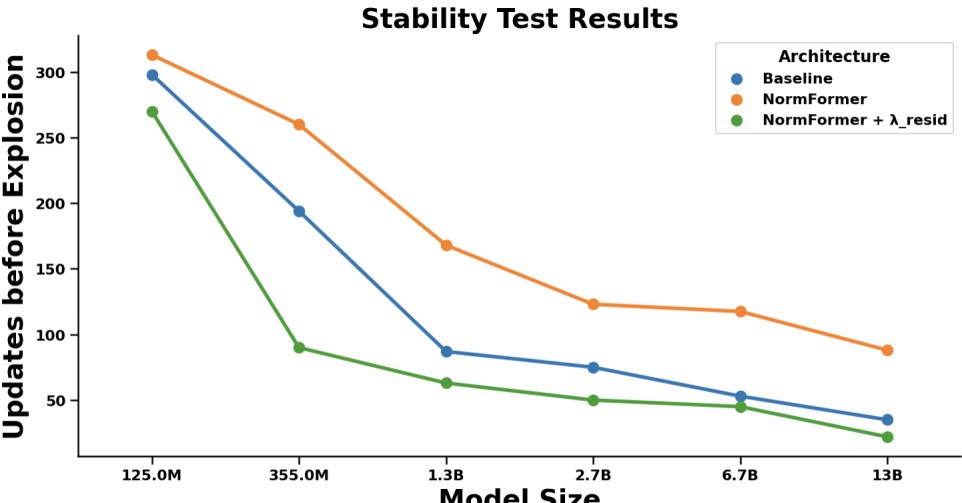

Figure 5: LR Stability Test: learning rate starts from 0 and linearly increases by `5e-5` at each training step until training destabilizes. NormFormer reaches a higher learning rate before destabilizing. Each data point is the median of 3 runs with a different random seed.

the attention outputs allows NormFormer to avoid this issue and remain stable with larger learning rates. Figure 5 also shows that $\lambda_{resid}$ reduces the stability improvement at all sizes.

### 6.1 RESIDUAL SCALING

By comparing adjacent NormFormer-125M and NormFormer-355M rows in Table 2 we can see that adding `ResScale` to `NormFormer` improves perplexity and zero shot performance for small scale CLMs. For 125M parameter MLM, `ResScale` improves pre-training perplexity marginally, but hurts fine-tuned performance. At 1.3 billion parameter scale, however, adding `ResScale` to `NormFormer` does not improve performance (Figure 2). Although it's not included in our tables, we find that `ResScale` without NormFormer is stronger than the baseline at small scale, but not large scale. This suggests that the negative result is caused by scale, rather than interaction with `NormFormer`.

Figure 6 in the appendix shows the average $\lambda_{resid}$ weights at each layer of different sized CLMs. We can see that at 125M and 355M parameters, the weights in the later layers are lower, indicating down weighting of the residual connection, whereas at the largest scale, 1.3B, the weights are larger deeper into the network. Adding the $\lambda_{resid}$ parameters to the other (earlier) residual connection in each layer, or using a scalar instead of a vector for each $\lambda_{resid}$, does not fix the large scale issue, but hurts small scale performance marginally. Additionally, Table 8 shows that initializing the network to place no weight on the layer outputs, and all the weight on residual connections, as proposed in (Touvron et al., 2021), does not improve performance.

## 7 ABLATIONS

This section provides evidence that removing any of our additions to the transformer block degrades performance on language modeling tasks, and that our additions improve language modeling performance across a wide range of hyperparameter settings. Experiments use 125M parameter CLMs, and are run with the default hyperparameters given in Table 7 in the appendix for 470 V100 Hours (100,000 updates for the baseline) unless otherwise mentioned.

**Removing any of the added operations hurts performance** Table 4 shows that none of the four introduced operations can be removed without degrading performance. Rows 2-5 remove each operation one at a time. In all cases perplexity increases, with the removal of `HeadScale` being

| Architecture | Valid PPL |
|---|---|
| NormFormer+ResScale | **15.88** |
| - Post-Attn LN | 15.92 |
| - FFN LN | 16.14 |
| - Head Scale | 16.22 |
| - Res Scale | 16.20 |
| + 3 More LN | 15.88 |
| Baseline | 16.37 |

Table 4: 125M parameter Language Modeling Validation perplexities after 470 V100 Hours of pretraining. Removing any of our proposed additions degrades performance (Rows 2-5). Adding more normalization inside the Multi Headed Attention (Row 6) does not impact perplexity at a fixed number of updates, but reduces throughput such that the model can only complete 87,500 updates vs. 92,500 for Rows 1-5 and 100,000 for Row 7. Note that these PPL scores are not directly comparable to other tables – they use a different validation set.

the most damaging and the removal of the Post-Attn LN being the least damaging. In Row 6 (`+ 3 More LN`) we try to introduce more normalization inside self attention, applying LN to the query, key and value features in addition to our 3 other operations, for a total of 6 new operations. In this setting, every other parameterized operation inside the transformer layer is an LN. We find that this does not change perplexities at a fixed number of updates, but reduces training speed by another 5%. This result suggests that there is not much upside to adding even more normalization on top of `NormFormer`.

**Other Experiments** Table 8 in the appendix compares `NormFormer` CLMs to related architectural modifications, both in terms of the stability and pre-training perplexity at 1.3B parameter scale. Table 5 in the appendix shows language modeling perplexities for 7 different hyperparameter configurations at 125M parameter scale, separated by horizontal lines. `NormFormer` outperforms the baselines in all settings.

# 8 CONCLUSION

We identify a mismatch in the gradients of Pre-LN transformer weights: earlier layers receive much larger gradients than later layers, while the optimal scaling of residuals is larger at earlier layers than at later layers. We propose `NormFormer`, which alleviates these issues by adding 3 extra operations to each transformer layer. These modifications help the gradient mismatch for fully connected parameters and improve validation perplexity and downstream task performance for both causal and masked language models. None can be removed without degrading performance back towards the baseline, and adding more normalization – at least of the types we have tried – does not improve performance. Since NormFormer primarily addresses the gradient mismatch by increasing the gradients to the last FFN layers while decreasing the gradient magnitudes in other parts of the network, future work could examine whether all 3 operations need to be added to every layer. Additionally, the small computational overhead associated with NormFormer could be alleviated by fusing the FFN LN with the preceding fully connected layer, with or without the mean centering and bias, which do not appear to improve pretraining perplexity. In general, we have shown that adding small numbers of learnable parameters in the right places in our architectures can alleviate certain issues in current state of the art networks. Future work should ascertain if there are additional similarly efficient modifications that can bring gains, while helping us understand current deficiencies further.

# 9 APPENDIX

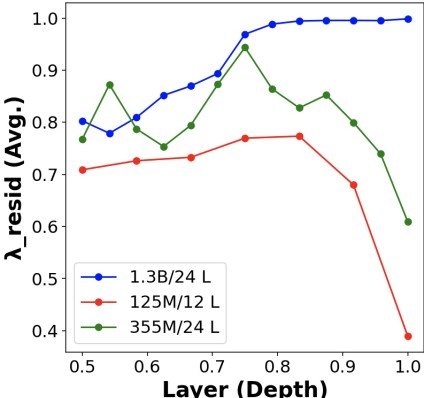

Figure 6: Average $\lambda_{resid}$ weights at each layer of different sized CLMs in the `NormFormer`+$\lambda_{resid}$ setting. Depth is layer number / total layers.

|  | Learning Rate | Setting Changes | Valid PPL |
|---|---|---|---|
| Baseline | 0.001 | - | 16.80 |
| NormFormer | 0.001 | - | **16.33** |
| Baseline | 0.003 | - | 16.37 |
| NormFormer | 0.003 | - | **15.88** |
| Baseline | 0.006 | - | 16.58 |
| NormFormer | 0.006 | - | **16.22** |
| Baseline | 0.003 | Longer Warmup | 16.50 |
| NormFormer | 0.003 | Longer Warmup | **16.06** |
| Baseline | 0.003 | GPT3 | 16.29 |
| NormFormer | 0.003 | GPT3 | **15.88** |
| Baseline | 0.003 | Clip Grad Norms at 0.1 | 16.46 |
| NormFormer | 0.003 | Clip Grad Norms at 0.1 | **16.14** |

Table 5: Longer Warmup: increase LR Warmup to 6,000 steps (from 500). GPT3: increase sequence length to 2048, increase dropout to 0.1, increase training budget to 1,000 V100 hours. Grad Clip: clip gradient norms at 0.1. NormFormer outperforms the baseline in all settings.

**Wikitext103** Table 6 shows that NormFormer can also provide gains on top of a well tuned language model in settings with much less data. We simply add our three operations to the architecture and hyperparameters of Baevski & Auli (2019). Convergence perplexity improves, and we reach the baseline perplexity in 70% as many steps. In this setting, `NormFormer` does not improve in the last 30% of training, which suggests that with more tuning the perplexity gap could be widened.

## 9.1 COMPARISON TO RELATED WORK

**Understanding Table 8** The two right most columns of Table 8 contains describes the result of two experiments from the same 1.3B parameter architecture: `Stability` indicates how many steps the experiment survived in the "LR Stability Test", where we increase LR from 0 to 0.1 linearly over 1,000 steps and `Perf`, where available, describes the validation perplexity of the model trained for 64 A100 days with sequence length 512. The table is sorted by Stability, with more stable configurations lower in the table. The left columns of the table indicate configuration. Row 14, which has completely empty configuration columns, is the baseline. The final row is `NormFormer`.

|            | Steps to Target PPL | Final PPL | A100 Hours |
|------------|---------------------|-----------|------------|
| Baseline   | 279.893             | 18.70     | 288        |
| NormFormer | 223.904             | 18.65     | 237        |

Table 6: Wikitext 103 results following Baevski & Auli (2019). `Steps to Target PPL`: at what percentage of the 280K steps did the model reach 18.70 perplexity. `Final PPL`: Best Perplexity
`.A100 Hours` Cost of reaching Target PPL.

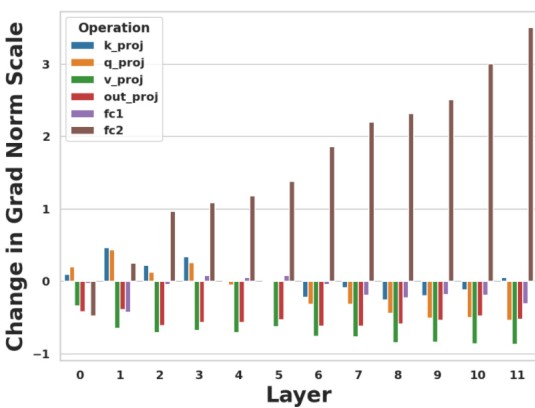

Figure 7: Change in grad norm with each operation of NormFormer compared to the baseline. Norms are the average between step 950 and 1000, normalized to control for different losses. 2.0 on the Y axis means the gradient to a parameter is twice as large as the baseline, on average. The NormFormer increases the norm to fully connected parameters in later layers, while reducing the gradient norm to attention parameters at all layers. The results are discussed in detail in Section 6.

The Architecture and LN Variant columns show whether we changed the model architecture completely and/or used a non-standard LayerNorm algorithm.

1. CogView[1] (Ding et al., 2021): implement all of the changes proposed in Section 2.4: (1) reduce the gradients to the embeddings by a factor of 10, (2) add an LN after attention, (3) add an LN after the second fully connected layer (not the first, like NormFormer) (4) change the layer norm formula to $LN(\frac{X}{Max(X)})$. `fairseq` already uses the attention score stabilization trick by default. In rows 19 and 25 where `Arch` is blank, we use the proposed LayerNorm formula but none of the other changes.

2. CatFormer[2] (Davis et al., 2021): Since no code was released, we re-implement CatFormer and set $\epsilon = 2$ and =448 to match 1.3B parameters. We guessed that the number of attention heads should be fixed in each layer and that LN positioning should not move, but these details are not clear from the paper.

3. LayerScale[3] (Touvron et al., 2021): We use the layer scale formulation from Section 2, which is similar to the $\lambda_{resid}$ discussed earlier, but with a weight on each layer's contribution to the main branch initialized at 1e-5, instead of a weight on the residual's input to the main branch initialized at 1.

4. FFNGeglu[4] (Shazeer, 2020): replace the FFN LayerNorm with 'FFNGeglu'. Although this is proposed as an activation function, it is also equivalent to just using the $\gamma$ of LayerNorm, with no normalization or bias.

5. T5[5] (Raffel et al., 2020): Switch LNs to the T5 variant, which removes the mean centering and bias.

6. PowerNorm[6] (Shen et al., 2020): Switch LN to PowerNorm, which is a variant of BatchNorm that shows promising results for smaller NLP models.

7. DeepInit[7] (Zhang et al., 2019): Multiply the initialization of each weight parameter by a factor of $\frac{1}{\sqrt{l}}$, where $l$ is the layer number starting from 1.

| | |
|---|---|
| Learning Rate | 0.003 |
| Batch Size | 524K Tokens |
| Parameters | 124M+ |
| Layers | 12 |
| Layer Dimension | 768 |
| Dropout | 0 |
| LR Warmup Updates | 500 |
| LR Scheduler | Linear Decay |
| Sequence Length | 1024 |
| Train Budget | 470 V100 Hours |

Table 7: Hyperparameters for ablations in Tables 4 and 7. This train budget allows the baseline model to run for 100,000 updates.

| ID | Arch. | Scale FC | Scale Attn | $\lambda_{Resid}$ | Scale Heads | LN Variant | Stability | Perf |
|---|---|---|---|---|---|---|---|---|
| 0 | - | ✓ | ✓ | | ✓ | PowerNorm[6] | 15 | - |
| 1 | CatFormer[2] | | | | | - | 34 | - |
| 2 | - | | ✓ | | | - | 52 | - |
| 3 | DeepInit[7] | | | | | - | 52 | - |
| 4 | DeepInit[7] | ✓ | ✓ | ✓ | ✓ | - | 58 | - |
| 5 | - | ✓ | ✓ | | ✓ | FFNGeglu[4] | 67 | 17.1 |
| 6 | - | | ✓ | | | - | 69 | - |
| 7 | LayerScale[3] | | | ✓ | | - | 76 | 17.5 |
| 8 | - | ✓ | ✓ | ✓ | ✓ | - | 76 | 17.1 |
| 9 | - | | ✓ | | ✓ | - | 81 | 17.1 |
| 10 | CogView[1] | ✓ | | | | CogView[1] | 86 | - |
| 11 | - | | | | ✓ | - | 91 | - |
| 12 | - | ✓ | | | | T5[5] | 93 | - |
| 13 | - | ✓ | | | | FFNGeglu[4] | 94 | - |
| 14 | - | | | | | - | 94 | 17.1 |
| 15 | - | ✓ | | | | No Bias or $\gamma$ | 95 | - |
| 16 | - | ✓ | | | | No Bias | 96 | - |
| 17 | - | ✓ | | | | No $\gamma$ | 98 | - |
| 18 | - | ✓ | | | ✓ | - | 112 | 16.8 |
| 19 | CogView[1] | ✓ | ✓ | | ✓ | CogView[1] | 118 | 17.1 |
| 20 | - | ✓ | | | | CogView[1] | 122 | - |
| 21 | CogView[1] | ✓ | ✓ | | | CogView[1] | 124 | - |
| 22 | - | ✓ | | | | - | 128 | - |
| 23 | - | ✓ | | | | PowerNorm[6] | 135 | - |
| 24 | LayerScale[3] | ✓ | ✓ | ✓ | ✓ | - | 148 | - |
| 25 | - | ✓ | ✓ | | ✓ | No $\gamma$ | 157 | - |
| 26 | - | ✓ | ✓ | | ✓ | CogView[1] | 165 | - |
| 27 | - | ✓ | ✓ | | ✓ | No Bias or $\gamma$ | 178 | 16.9 |
| 28 | - | ✓ | ✓ | | ✓ | No Bias | 184 | - |
| 29 | - | ✓ | ✓ | | ✓ | T5[5] | 189 | - |
| 30 | - | ✓ | ✓ | | | - | 192 | 16.7 |
| 31 | - | ✓ | ✓ | | ✓ | - | 200 | 16.6 |

Table 8: Stability and Performance for different architectures for 1.3B parameter CLMs, see Section 9.1 for details.

8. No $\gamma$: Freeze $\gamma = 1$ in LN
9. No Bias or $\gamma$: Freeze $\gamma = 1$, bias=0 in LN
10. No Bias: Freeze bias=0 in LN.

**Results** The results suggest that other proposals to mitigate instability in other domains, like vision, text to image generation and reinforcement learning do not improve stability or pre-training performance in the CLM setting.

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
