# OpenReview forum: "NormFormer: Improved Transformer Pretraining with Extra Normalization"
_ICLR.cc/2022/Conference — ICLR 2022 Submitted_

### Official Review · Reviewer_uRwh · 2021-11-01

**Correctness:** 4
**Technical Novelty And Significance:** 2
**Empirical Novelty And Significance:** 2
**Recommendation:** 5
**Confidence:** 4

**Main Review:**

Strengths
1. The paper is well-written and easy to follow.
2. The four operations are easy to implement and applicable to many tasks using transformer architectures.
Some visualizations, e.g., Figure 3, provide straightforward comparisons.

Weaknesses
1. Although the four operations are shown effective in accelerating pretraining convergence or boosting downstream performance, they look more likely engineering tricks. These tricks have been widely used in prior works, and I don't see much insight in being used here.
2. There is a lack of analysis of why the gradient mismatch issue exists in Pre-LayerNorm transformers and why the proposed four operations can alleviate it. The paper mainly focuses on what it does rather than justifying why doing that.
3. There are two scaling operations added in Figure 1, but three are plotted in Figure 4. It is unclear how the left two subfigures in Figure 4 correspond to the scaled residual connection in Figure 1.

**Summary Of The Paper:**

This paper aims to improve pretraining Pre-LayerNorm transformers by alleviating two issues: early layers have much larger gradients than later ones, and naive residual learning can't provide optimal weighting. To this end, it proposes to add two LayerNorms after the multi-head attention and the GELU non-linear activation in FFN, respectively. It also adds learnable scaling coefficients for the FFN residual and the attention head outputs. The four modifications are applied to both casual and masked language modeling with improvements observed in downstream tasks.

**Summary Of The Review:**

In summary, this paper proposes to add four operations, two LayerNorms and two scaling parameters, in the Pre-LN transformer layers. The operations are simple and can help stabilize pretraining and improve downstream performance in several NLP tasks. However, I think the method novelty is not enough. Moreover, the paper doesn't provide convincing motivation why the four operations help handle the issues.

---

> ### Author Response · Authors · 2021-11-19
> **Novelty and Related Work**
>
> The contribution of this paper is:
> 1) Propose out a small set of architectural modifications that improve performance at both small and large scale.
> 2) Test Related Work and many baselines since there are so many other possible architectural modifications, as you mention.
> 3) Try to understand why the modifications improve performance.
>
> We have done a good job with 1 and 2 and made some progress on 3.
>
> ### Novelty
> We simplified our proposal to only add 3 operations: 2 layernorms and scaling of attention heads, and references to 11 pieces of Related Work in Section 3.
>
> (1) Most Related Work focuses on scaling residual connections, which is no longer part of our proposal.
> (2) We have not found any related work that puts layernorm between the FFN layers, the most important of our additions.
> (3) Most importantly, as we show in the newly added Table 8, as well as Table 2, none of the Related Work performs nearly as well as NormFormer.
>
> ### Analysis
>
> > There is a lack of analysis of why the two issues exist in Pre-LayerNorm transformers and why the proposed four operations can alleviate them.
>
> I'm not sure what two issues you describe. The issue we are tackling is that the gradients to layer 0 are larger than to later layers: gradient mismatch.
> This prevents various baselines from using a large enough learning rate, because, as shown in (Table 2, Figure 5, Table 8) they explode.
> NormFormer (specifically the LayerNorms) alleviates this by reducing the gradient norm to early layers without reducing the gradient norm to later layers, thereby allowing faster training (more learning per step).
> We do not know why this happens theoretically, but we think it is useful to the field for us to identify this issue, even if we aren't the ones to do the theoretical work of understanding the root cause.
>
>
> ### Figure 1 and 4 inconsistency
> We fixed Figure 1 to be consistent with Figure 4 (all now show 3 modifications).
> Residual Connections are discussed in 6.1 plotted in the appendix.
>
>
> We'd be happy to perform further analysis over the weekend if there are questions you want answered.
> We've also included learning rate matched and unmatched baselines in Table 2, extensive stability tests, and at 2.7B parameters in the updated version.

---

> > ### Comment · Reviewer_uRwh · 2021-12-02
> > **Need to bridge two logical gaps**
> >
> > Thanks for fixing the figures and clarifying the novelty. From the perspective of empirical results and practical values, the proposed FFN LayerNorm, Post Attn LayerNorm, and HeadScale are promising. But from the paper's storytelling perspective, they are not well-motivated, also pointed out by reviewer ust2.
> >
> > The paper starts from the gradient mismatch issue and introduces two norms and one scaling. However, the experiments mainly show how they can lower pretraining perplexity, improve zero-shot accuracy, and boost fine-tuned performance. Figure 3 is better to place at the beginning of experiments since this experiment is directly related to the gradient mismatch issue. There is a lack of analysis of why reduced gradient mismatch can improve model generalization. I notice that the authors mentioned some intuition for why reducing gradient mismatch helps in the responses to reviewers ust2 and jDo1. I think more investigations on this intuition can serve as part of the motivation. Basically, the paper needs to address the logical gap between mitigating gradient mismatch and improving generalization.
> >
> > Another logical gap is why the proposed three components help reduce gradient mismatch. I realize that giving a theoretical analysis of why the two norms and one scale can reduce gradient mismatch may be difficult. But some more in-depth ablation studies and hypotheses are necessary. Otherwise, it is difficult for readers to learn knowledge here and use them somewhere else.
> >
> > BTY, Figure 5 looks too big, which may not fit the rest of the paper. For me, the stability test experiments' setup is unclear. I see the authors mention something like " linearly increase lr to 0.1 (way too high) over 1000 steps " in the responses to reviewers ust2 and jDo1. But I can't find/infer this information in Figure 5 and the paragraph describing it. I think a straightforward experiment setup is necessary.
> >
> > Overall, I appreciate the authors' efforts in addressing my questions. But I think this paper should improve its motivation and experiments design to make the whole logic smoother. Nevertheless, I decided to increase the score to 5 considering other reviewers' comments.

---

### Official Review · Reviewer_ust2 · 2021-11-02

**Correctness:** 3
**Technical Novelty And Significance:** 3
**Empirical Novelty And Significance:** 4
**Recommendation:** 5
**Confidence:** 4

**Main Review:**

This paper ​proposes some modifications including two normalizations and two new scaling operations to mitigate the issue of gradient magnitude mismatch inside the transformer. The resulting Normformer improves pretraining perplexity and downstream task performance with a negligible increase of computing cost.

(1) The paper is concisely written and easy to understand. However, the contributions are not clearly presented in the Introduction. Besides, since the section of related work is put at the end of the paper, the readers may get confused about how the proposed method differentiates from existing works.

(2) The issue of gradient magnitude mismatch is not well explained. For example, how the gradient magnitude mismatch affects the training. Moreover, how the optimal weighting of residuals is connected with this issue is unclear. As they are the main motivations of the proposed method, it is suggested to provide a detailed explanation about how gradient magnitude mismatch, optimal weighting connects with the proposed method.

(3) As shown in Fig.3 of  [1], the PreLN with warmup can effectively tackle the problem of gradient magnitude mismatch inside the FFN of the transformer. Can you explain why the issue still emerges in transformers with a common learning schedule? For example, can you provide some intuitions on why gradients at early layers are much larger than at later layers in PreLN transformers? As shown in Fig.3, gradnorm in only several layers are visualized.

(4) I appreciate that the ablation study is conducted to show the performance gain of each component in  NormFormer. However, since four modifications are proposed to solve a single issue, it is expected to see how these four modifications affect the gradient magnitude and how these modifications interact with each other.

(5) The effect of HeadScale parameters is overclaimed. The readers cannot see any evidence showing that HeadScale parameters can adjust the importance of attention heads.

(6) Are there other studies investigating training instability? Please discuss more in related work. It is good to show the effectiveness of NormFormer through extensive experiments.

(7) The results seem convincing and competitive.


**Summary Of The Paper:**

Competitive empirical results but four modifications (two norms and two scalings) are not well motivated and explained.

**Summary Of The Review:**


Overall, this paper proposes NormFormer to improve the PreLN transformer by introducing some modifications. The proposed techniques are empirically successful but not well motivated. I hope that the authors can firstly present the issue inside the PreLN transformer clearly and then provide a detailed discussion about why the proposed modifications can mitigate this issue.

---

> ### Author Response · Authors · 2021-11-19
> **Added Related Work+ Ablations**
>
> Thanks for the review, it caused a lot of changes!
> I've tried to improve and motivate the proposed modifications in the updated version, as well as testing NormFormer at 2.7B CLM scale.
> Let us know if there is anything else that would make it clearer!
>
> ### (1, 4, 6)
> We moved Related Work to the beginning of the paper, and tried to clarify our contribution in the introduction.
> We added and implemented a lot of the methods' impact on stability in Table 8.
> We've also repeated the ablation on all combinations of the 4 proposed modifications in Table 8 (Stability test.)
>
> ### (2, 3) Gradient Norm Mismatch
>
> I can't find any theoretical justification for why gradient norms are higher at layer 0 for pre-ln than later layers.
> [Xiong et. al](https://arxiv.org/pdf/2002.04745.pdf) figure 3 shows a bit of this empirically, but their theory concludes only that
> "the scale **almost** keeps the same for different layers in the Pre-LN Transformer".
> I have consistently seen this pattern empirically at a wide variety of scales and would be happy to make another table about it.
>
> My intuition for why reducing gradient mismatch helps is as follows:
>
> In our experiments, increasing LR has led to two outcomes: (1) performance improves or (2) gradients explode (as in Table 2 2.7B baseline, and the stability test).
> Our learning rate is limited by the largest gradient, but the amount we update our parameters per step is a function of the average gradient.
> By reducing the largest gradients without reducing the smallest gradients (aka reducing mismatch), we are able to train at higher learning rates and still make the same amount of progress per step.
>
> In the updated version, we've tried to show that the effect is meaningful by including more stability test results.
> In this test we linearly increase lr to 0.1 (way too high) over 1000 steps and see how long an architecture can survive before gradients explode.
> NormFormer consistently lasts longer than the baseline, and the gap increases with scale.
>
> This isn't proof that reducing gradient mismatch causes better results, but that NormFormer causes more tolerance of a high learning rate, which is a step in the right direction.
>
> (2b) Optimal weighting of residuals is connected in related work, but it doesn't seem to help as much as the other modifications, so we've removed it from our proposed architecture.
>
> ### (5) Attention Heads
> The Head Scale parameters help performance (as shown in Table 4), and vary significantly within each layer (Figure 4).
> This suggests that they are increasing/decreasing the importance of attention heads.
> If there is another piece of evidence you'd be interested in seeing I'd be happy to run more experiments!

---

> > ### Author Response · Authors · 2021-11-29
> > **Any Questions?**
> >
> > Let us know if you have any other questions.

---

> > > ### Comment · Reviewer_ust2 · 2021-11-29
> > > **Motivation is not sufficiently strong**
> > >
> > > I appreciate the responses to my questions. The issue related to residual reweighting is well resolved as the authors put it to the related work. Moreover, the authors also provide a more detailed explanation for the relationship between GradNorm mismatch and models' performance. I also recognize that the experimental results are competitive and exciting. However, my major concern is still unsolved. I would like to suggest the authors provide sufficient motivations to present why we need to use more LNs after MHA and after the first FC rather than other positions. It is important to let readers know why normalizing the output of MHA and the first FC layer is so crucial for tackling the issue of GradNorm mismatch. Hence, I would keep my initial ratings unchanged.

---

> > > > ### Author Response · Authors · 2021-11-29
> > > > **Motivation**
> > > >
> > > > In the newly added Table 9, we compare stability and performance to the architecture proposed in CogView, which puts the FFN_layernorm after fc2 and uses the post MHA layer norm.
> > > > In table 4 and table 9, we show that removing either of the layer norms degrades performance and stability.
> > > > So, in short, we show empirically that the changes are necesarry and try to understand why in the analysis section.
> > > > We can't prove that our changes are crucial for tackling grad norm mismatch, but we have shown (1) our changes reduce grad norm mismatch, (2) our changes improve stability and performance (3) removing any of our proposed modifications, or moving them to different positions degrades stability and performance.

---

### Official Review · Reviewer_jDo1 · 2021-11-04

**Correctness:** 3
**Technical Novelty And Significance:** 3
**Empirical Novelty And Significance:** 3
**Recommendation:** 8
**Confidence:** 5

**Main Review:**

I think this is mostly solid work. The writing and methods are easy to understand and the contributions are clear and convincing. However I think the baselines could be more challenging and the analyses more convincing.

# Feedback

I appreciate the authors' clear communication and presentation of their work. I also applaud their use of GPU-hour-matched baselines.

My main concern with the present work is that it is evaluated against baselines with no stability/trainability interventions. Such interventions exist; the authors mention a number of them in the *Related Work* section. But without including any of them, it's difficult to directly assess whether the present work is an improvement over existing methods.

I’m a bit confused. The caption of Figure 2 says “The green star shows the point where NormFormer outperforms the baseline’s lowest perplexity, reflecting 22% and 43% compute savings for CLM and MLM models, respectively.”, while the first paragraph of results says “NormFormer matches Pre-LN models while needing only 60% and 57% as much compute for CLM and MLM models, respectively”. Either I’m misunderstanding your metrics, or one of these statements is incorrect. Regardless, I would encourage the authors to pick one efficiency metric and use it consistently and exclusively. Personally, I think “x% as much compute” or “x% the amount of compute” are more intuitive, as they clearly mean “(normformer/baseline)*100 amount of compute”.

I don’t find the gradient norm analysis (Section 4.1) particularly convincing. It’s hard to know whether the effect of NormFormer on gradient norms is meaningful, and even whether it's causally related to the performance improvements. It would be great to see some experiments demonstrating the causal effect of gradient norm changes on trainability. At the very least I think readers would appreciate some theoretical justification or intuition. Why is it desirable to have similar gradient norms across layers? Also in Figure 3, most of the Layer 0 gradient norms are out of the frame in both Pre-LN plots, making comparison difficult. I suggest changing all three plots to a logarithmic y-axis.

I think it would be very informative to examine the scaling parameters over the course of training—especially early in training. The early phase of transformer training seems to be the most fraught. This is when the loss tends to explode, and this is when you (and others) show the largest changes in gradient magnitudes. Analyzing only the final learned parameter values ignores the most important part of the story.

There seems to be an implicit assumption that the converged scaling parameters (e.g. Figure 4) are “good for training”. Does initializing the NormFormer parameters to be of similar magnitude to what is observed at convergence yield improvements over the current initialization scheme?

LayerNorm consists of a scale and bias, but only the scale parameter is analyzed. It’s possible that the bias could counteract the effect of the scaling. Although I think this scenario is unlikely, it should nevertheless be accounted for.

The ResScale results (end of Section 4) should be expanded and plotted. And why did the authors choose only to look at the minimum value of lambda_resid? The layerwise analysis of lambda_resid should be repeated with different metrics.

Please include quantitative results from the **Other Experiments** section.

There are a few relevant papers that might be worth discussing (or at least referencing). Zhang et al. conducted earlier research on the importance of initialization (*Improving Deep Transformer with Depth-Scaled Initialization and Merged Attention*, 2019). GradInit (Zhu et al., 2021) is recent, and also quite relevant. PowerNorm (Shen et al., 2020) and Catformer (Davis et al., 2021) both also take similar approaches to solving transformers’ trainability issues. LayerScale (from Touvron et al.’s *Going Deeper with Image Transformers*, 2021) examines the trainability problem in vision transformers and find a solution that appears similar to one the present innovations. Finally, Brock et al. (*Characterizing signal propagation to close the performance gap in unnormalized ResNets*, 2021) examine CNNs, but their approach and findings are quite relevant.


**Summary Of The Paper:**

NormFormer improves on Pre-LN transformers by making the following modifications: learnable scaling parameters for each dimension of the output of each attention head prior to concatenation across heads (*Scaled Attention*); layer norm on the attention output (*Post Attn LN*); layer norm on the FFN nonlinearity output (*FFN LN*); and learnable scaling parameters for each dimension of the skip connection around the FFN (*Scaled Residuals*). They apply NormFormer to GPT3- and RoBERTa-style model configurations, and find that NormFormer models reach baseline iso-accuracy in 22%-43% less time, and achieve notably lower (higher) iso-time perplexity (accuracy). They then conduct a number of analyses to attempt to understand why NormFormer works.

**Summary Of The Review:**

Transformers are notoriously unstable to train. Progress on this problem is of clear value to the field. The baselines could be more challenging and the analyses more convincing, but if these issues are addressed—which I am optimistic about—I think the paper could be a meaningful contribution to the field.

---

> ### Author Response · Authors · 2021-11-19
> **Implemented, Tested,Cited more Related Work+ 2.7B scale**
>
> Thanks for the careful, thought provoking review.
> > Related Work + Other Experiments Table + Baselines
>
> Thank you so much for these, I hadn't seen any of the papers you mentioned! We now cite them all and include results for the stability test and a few larger runs at 1.3B params in Table 8 in the appendix.
> None of the variants improve over NormFormer, but it does seem like mean centering and bias in LayerNorm can be removed without much harm. We've also included learning rate matched and unmatched baselines in Table 2, as well as results at 2.7B parameters.
> Additionally,
> I'd be happy to train anything else over the next few days if there is a particular other baseline number you'd be interested in.  Reading these papers also led us to remove some language that suggested that scaling residual connections is part of our contribution. It also seems to be less valuable as depth increases.
>
> > Consistent efficiency metric + Figure 2 confusion
>
> Good point. Switching to Cost="% of compute required to reach the baseline perplexity". Figure 2 stars now reflect this.
> In short, at 1.3B Params we match GPT-3-Large published performance at 62% of the cost, and our stronger high learning rate baseline at 79% of the cost.
>
> > Does initializing the NormFormer parameters to be of similar magnitude to what is observed at convergence yield improvements over the current initialization scheme?
>
> I tried this at 1.3B parameters and trained 10% of the way and it doesn't seem to help. LayerScale tried this experiment and got the same result, which I find somewhat surprising.
>
> > LayerNorm consists of a scale and bias, but only the scale parameter is analyzed. It’s possible that the bias could counteract the effect of the scaling. Although I think this scenario is unlikely, it should nevertheless be accounted for.
>
> Table 9 now includes No bias and Just Bias. As you predicted, it appears that the bias does not do very much. Unfortunately, we don't save a noticeable amount of compute by removing the bias, so to keep our proposal simple we keep it in.
>
> > Examine the scaling parameters over the course of training—especially early in training.
>
> [Here](https://user-images.githubusercontent.com/6045025/142498257-cd635739-bcae-460c-9143-afd6f01a959f.png) are the scaling parameters (averaged at each layer) for a 1.3B CLM NormFormer model trained for 80K steps. I think it coheres with the message of Figures 3 and 4:
> FFN LayerNorm is downscaling earlier layers more than later layers, thereby reducing mismatch. We don't see a consistent pattern in the other two scalers.
>
> > Why did the authors choose only to look at the minimum value of lambda_resid?
>
> Great catch, just a mistake. In the new version we've repeated the analysis at more scales using mean instead of min.
>
> ### Grad Norm Analysis
> I can't find any theoretical justification for why gradient norms are higher at layer 0 for pre-ln than later layers.
> [Xiong et. al](https://arxiv.org/pdf/2002.04745.pdf) figure 3 shows a bit of this empirically, but their theory concludes only that
> "the scale **almost** keeps the same for different layers in the Pre-LN Transformer".
> I have consistently seen this pattern empirically at a wide variety of scales and would be happy to make another table about it.
>
>
> My intuition is basically as follows:
> In our experiments, increasing LR has led to two outcomes: (1) performance improves or (2) gradients explode (as in Table 2 2.7B baseline, and the stability test).
> Our learning rate is limited by the largest gradient, but the amount we update our parameters per step is a function of the average gradient.
> By reducing the largest gradients without reducing the smallest gradients (aka reducing mismatch), we are able to train at higher learning rates and still make the same amount of progress per step.
>
> In the updated version, we've tried to show that the effect is meaningful by including more stability test results (more scales, more baselines).
> In this test we linearly increase lr to 0.1 (way too high) over 1000 steps and see how long an architecture can survive before gradients explode.
> NormFormer consistently lasts longer than the baseline, and the gap increases with scale.
>
> This isn't proof that reducing gradient mismatch causes better results, but that NormFormer causes more tolerance of a high learning rate (which means it has a lower max gradient). If we combine that evidence with the performance improvements in Table 2 (+ the fact that a 2.7B learning rate matched baseline exploded, + the fact that removing the layer norms degrades performance and stability (Table 9) it seems reasonable to conclude that at least part of the reason NormFormer outperforms is reducing gradient mismatch.
>
> I've added some language to the introduction to try to make this clearer, but your ideas are very welcome!

---

> > ### Comment · Reviewer_jDo1 · 2021-11-20
> > **Very interesting!**
> >
> > Thanks for your detailed response. I really appreciate the additional results and explanation. They make the work a lot more convincing.
> >
> > I think the points you make about the gradient norm analysis are interesting, though speculative. I think it could be quite valuable to perform additional analyses that test your ideas, but I don't consider it necessary for the viability of this work. I do think that at the very least you could formalize your statement a bit and toss it into the appendix.
> >
> > Unless other reviewers are able to convince me of a fatal flaw in this work, I will recommend it be accepted, and have updated my score from a 5 to an 8 accordingly.
> >
> > Two minor issues:
> > **Section 9.1 Comparison to Related Work. Understanding Table 8**
> > You say “Row 15, which has completely empty configuration columns, is the baseline.” Row 15 appears to have values in the configuration columns. Did you mean Row 14?
> >
> > Given that NormFormer increases the time per training step by 2-6%, I think the comparison in Table 6 would be fairer if it used wall-clock-time or some other metric that controls for the increased computational burden per NormFormer step.

---

> > > ### Author Response · Authors · 2021-11-22
> > > **Fixed minor stuff**
> > >
> > > Fixed minor stuff, thanks.

---

### Official Review · Reviewer_o2vf · 2021-11-06

**Correctness:** 4
**Technical Novelty And Significance:** 3
**Empirical Novelty And Significance:** 3
**Recommendation:** 8
**Confidence:** 4

**Main Review:**

In general, this is a good paper with sufficient and exhaustive experiment results. The 60% time-saving is impressive. The ablation studies show that both rescaling and more layer normalizations are contributing to better quality in MaskLM accuracies and zero-shot tasks. The 3 GPT-3 models of different size scales prove the generalizability of the techniques.

There are some small points that I think the authors can improve to make their statement stronger:
* The current framework, though claimed to be Transformer-related modification, is conducted in GPT-3 only. It would be helpful to see whether this Transformer modification can be used in other frameworks such as BERT and T5 or a single Transformer model.
* The Peak LR experiment does show that the current modification will allow the NormFormer to have up to 0.0275 learning rate stability, which compared to the original 0.02 baseline does not have a magnitude difference. It would not be that a strong statement to argue for the added robustness if higher LR would not always lead to better quality in general. I hope the authors can show more evidence if this is the case and the reason why they care about this aspect.


**Summary Of The Paper:**

This paper proposes a framework that adds extra layer normalizations and scaled residual layer to the normal GPT-3 model. Empirical results demonstrate faster convergence speed and its robustness towards the learning rate.


**Summary Of The Review:**

The paper shows a simple yet effective way of modifying the existing GPT-3 framework by increasing only a small number of parameters and computation per epoch, which can be a potential addon to many existing Transformer architectures.

---

> ### Author Response · Authors · 2021-11-19
> **RoBERTa Results + Improved Stability Test**
>
> Thanks for the review, on your two points:
>
> In cased you missed it, we show results applying the modification to Roberta, which is similar to Bert, in Table 2.
> In the new version, we have improved and expanded our Stability Test  to include different model sizes and many more baselines. We now measure “Number of updates survived” instead of maximum LR which shows that NormFormer often survives >2x longer than all baselines.
>
> We also added a table (and some sentences) showing why we care about allowing higher LR: it allows training to a target perplexity in fewer steps. In our experiments, increasing LR has led to two outcomes: (1) performance improves or (2) gradients explode (as in Table 2.7B and the stability test results). One major of advantage of NormFormer is that it allows training at higher learning rate.
>
> We’ve also added  CLM results at 2.7 Billion parameter scale, RoBERTa results with and without residual scaling, and more detailed comparison to related work.

---

> > ### Author Response · Authors · 2021-11-29
> > **Any questions?**
> >
> > Let us know if you have any other questions.

---

### Decision · Program_Chairs · 2022-01-20

**Decision:**

Reject

**Comment:**

This submission proposes a few small changes to the (PreLN) Transformer architecture that enable training with higher learning rates (and therefore can result in faster convergence). The changes include the addition of two layer norm operations as well as a learnable head scaling operation in multi-headed attention. The proposed operations add only a small computational overhead and should be simple to implement. Experiments are conducted on language modeling and masked language modeling, with improved results demonstrated at various scales and according to various evaluation procedures. The paper also includes a good amount of ablation study as well as some analysis. Reviews on the paper were mixed, and a great deal of changes were made to the paper during the rebuttal period. To summarize the concerns and recommendations, reviewers requested
- better connection between the proposed changes and the purported issue (gradient scale mismatch between early/late layers)
- better analysis of why gradient scale mismatch is a major issue and investigation of where it comes from
- better comparison to existing techniques that allow for higher learning rate training of Transformers
- additional experiments on different model types and ideally different codebases/implementations

I think overall this is a solid submission, since it proposes a simple change that is reasonably likely to be helpful (or at least not harmful). However, I think that there are enough concerns with the current draft and there were enough changes made during rebuttal that this paper should be resubmitted to a future conference. I would suggest the authors take the final updated form from this round, add additional motivation/analysis/experiments, and resubmit, and I suspect a positive outcome.